# Vaccination against misinformation: The inoculation technique reduces the continued influence effect

**Klara Austeja Buczel, Paulina D. Szyszka, Adam Siwiak, Malwina Szpitalak, Romuald Polczyk** [ORCID]*

Institute of Psychology, Jagiellonian University, Kraków, Poland

* romuald.polczyk@uj.edu.pl

**Data Availability Statement:** All relevant data are within the paper and its Supporting Information files.

## Abstract

The continued influence effect of misinformation (CIE) is a phenomenon in which certain information, although retracted and corrected, still has an impact on event reporting, reasoning, inference, and decisions. The main goal of this paper is to investigate to what extent this effect can be reduced using the procedure of inoculation and how it can be moderated by the reliability of corrections' sources. The results show that the reliability of corrections' sources did not affect their processing when participants were not inoculated. However, inoculated participants relied on misinformation less when the correction came from a highly credible source. For this source condition, as a result of inoculation, a significant increase in belief in retraction, as well as a decrease in belief in misinformation was also found. Contrary to previous reports, belief in misinformation rather than belief in retraction predicted reliance on misinformation. These findings are of both great practical importance as certain boundary conditions for inoculation efficiency have been discovered to reduce the impact of the continued influence of misinformation, and theoretical, as they provide insight into the mechanisms behind CIE. The results were interpreted in terms of existing CIE theories as well as within the *remembering framework*, which describes the conversion from memory traces to behavioral manifestations of memory.

## Introduction

### Continued influence effect (CIE)

There are numerous cases where a certain belief persists despite being proven wrong. The widespread myth that vaccines cause autism in children is one of the most well-known examples of this phenomenon [1]. Misinformation has become a significant concern in contemporary society [2]; the strategies used to oppose it do not deliver expected results and the ongoing research suggests that corrections are not sufficient in reducing the effects of misinformation [3–5].

In psychology, the influence of retracted information on reasoning and decisions is called the continued influence effect (CIE) [5–7]. In a typical CIE procedure, participants are being presented a scenario about some fictional event (e.g. a warehouse fire, jewelry theft, or a bus accident) in which certain information is given and then retracted and/or corrected [8].

**Funding:** The authors received no specific funding for this work.

**Competing interests:** The authors have declared that no competing interests exist.

Although it might seem that the respondents, aware of the retraction, should formulate conclusions without misinformation, the retraction only slightly reduces the reliance on misinformation [8–18]. Sometimes, the retraction seems to be completely ineffective [6, 19] and, in some cases, it can even cause a paradoxical increase of reliance on misinformation [20]; see, however, e.g. [21, 22]. There is also a tendency that people believe more in misinformation than in corrections regardless of the sources of retractions [19, 23].

Two leading cognitive explanations of CIE have been proposed, which are not exclusive and may be complementary. Both are focused on memory mechanisms, assuming that some kind of memory error is responsible for the phenomenon. The first of the two main theoretical approaches assume failure in updating and integrating information in the mental models of unfolding events [6, 8, 24–27]. The mental model is defined as a representation based on one's available information and knowledge, built on the principle of cause-effect relationships [28]. Mental models are not static structures: they can be updated locally, where minor elements are added to the existing model, or globally, where the whole model is reconstructed and a new one is created [29, 30]. In the case of CIE, it is assumed that the model updating leads to the local consistency but conclusions are drawn globally, leading to errors due to local sustaining of misinformation [25]. Johnson & Seifert [6] proposed that misinformation is maintained in the model because of its causal role for the described event–when the information central to the model is invalidated, a gap arises that renders the model inconsistent. One of the possible solutions to this problem is keeping the discredited misinformation in the model because of its role in filling the gap. In other words one will prefer to keep the coherent but inaccurate model to the incoherent but accurate model; this consequently makes an answer that is consistent with misinformation better that no answer at all, even though it is potentially erroneous. Therefore, it is proposed to conceptualize CIE as a result of a failure in integrating information in the model when the retraction is processed [31–35].

The second theoretical approach argues that CIE arises from selective retrieval of misinformation from memory [10, 12, 17, 36–38]. According to this approach, both misinformation and its retraction are stored simultaneously in memory; consequently, CIE occurs when there is insufficient suppression of misinformation. One possible explanation within this approach is based on the distinction between automatic and strategic processes [39]. The former relies on the context-free recognition of information previously encountered, usually of a general nature, while the latter allows the extraction of additional content, e.g. context, source, and truthfulness of the information, and requires the use of cognitive resources. It is assumed [40] that individual parts of information in the memory compete for activation and that only those that win the competition are available. At the same time, associations between the information and their context (e.g. source) are not accessible and strategic processes must be used to gain access to them. Thus, CIE occurs when misinformation wins the competition in activation and is retrieved automatically, e.g. as a result of a presence of an appropriate clue; at the same time controlled processes are disrupted and fail to retrieve the correction [36]. This kind of automatic retrieval could rely on familiarity and fluency, thus leading lead to the illusory truth effect, where familiar (e.g. repeated) information is more likely to be perceived as true [41]; when misinformation or any clue connected with it appears, the chance to activate it increases. Alternatively, according to negation processing models, information being encoded is always initially treated as if it were true and is only subsequently falsified by attaching a negation tag [42]. However, negation tag retrieval uses cognitive resources, so when strategic processes fail, one can "lose" the negation tag, retrieving only the misinformation.

Besides memory factors, non-memory factors such as motivational factors (e.g. person's attitudes and the worldview) are also being taken into account in explaining CIE. It is proposed, for example, that motivated reasoning [43] is responsible for misinformation

compliance when misinformation relates to political views or prejudices [9, 11, 13, 20, 44]; however, see: [45–48]. Furthermore, more skeptical people tend to reject misinformation and accept retractions more correctly [49]. It is also possible that one does not accept the retraction because it seems more unreliable than misinformation [19, 23] or comes from an unreliable source [16, 19, 50]. Connor Desai et al. [51] even argued that if the misinformation comes from a source that is perceived to be more reliable than the source of the retraction, it may be considered rational to rely on misinformation. Recent results obtained by Susmann & Wegener [52] also indicate that motivational factors may play a key role along with cognitive factors even in situations when research material is purely neutral.

From a practical point of view, recognizing effective techniques for limiting the impact of misinformation in the CIE paradigm seems to be extremely important. As summarized in 2012 by Lewandowsky et al. [7], three factors that may increase the effectiveness of retractions were identified: warning against exposure to misinformation [10], repetition of the retraction [12], and an alternative to misinformation that fills the causal gap left by the retraction [6] (for countering non-laboratory misinformation, see, e.g.: [53]). It seems that the most effective technique is a combination of a warning and alternative, which reduces reliance on misinformation to a greater extent than its components alone [10]. Unfortunately, all these techniques seem problematic for several reasons. First, none of them can eliminate the impact of misinformation, and sometimes the alternative is not effective at all [23, 26]. Repetition of a retraction does not reduce CIE more than a single retraction when misinformation is presented once; however, the repetition of the retraction is more effective than a single retraction when misinformation is repeated several times [12]. The techniques mentioned are also subject to critique when it comes to their practical use. Finding a credible alternative in non-laboratory conditions is often impossible [7] and warning people before they encounter specific misinformation is rather unrealistic [54]. Therefore, it seems that further exploration of these factors is necessary, focusing especially on significant practical applications. In this study, we present an experiment that investigates the effect of inoculation on CIE.

## Inoculation theory

In the inoculation theory [55, 56] a metaphor of biological vaccination is used to illustrate a mechanism of immunization against persuasion: as in the case of an infection, people can be "vaccinated" against social influence by being exposed to a weaker kind of persuasive attack. Similar to the biological vaccination, the psychological inoculation procedure leads to producing "antibodies" in the form of counterarguments that can immunize an individual against subsequent persuasion and protect their attitudes from change [56].

The procedure of inoculation consists of two basic components: Warning and Refutational Preemption. The Warning makes it visible to the individual that there are possible arguments for changing their attitude, which produces the feeling of threat. According to the theory, to activate the motivational processes responsible for attitude reinforcement, a person going through a process of inoculation must feel that one of their belief is in danger. Without this feeling of threat, proper inoculation would be impossible [55–59]. In addition to the sense of threat, factors such as the involvement in defending an attitude [60], the emotion of anger [61], and the accessibility of the target attitude in one's memory [60, 62] play a crucial role in inoculating against persuasion. However, although necessary, the very sense of threat triggered by a warning is not sufficient for producing effective resistance to persuasion–a Refutational Preemption also has to be made, i.e. exposure to arguments against one's attitude and then refutation of these arguments, which strengthen one's attitude due to the production of counterarguments to the content of future persuasion [55, 58, 60, 61, 63].

Regardless of whether these counterarguments are the same or different that the content of the persuasive message, they can protect against persuasion [56, 58, 61, 62, 64]. The inoculation procedure can be processed actively (participants write an essay containing counterarguments) or passively (participants read counterarguments prepared by the experimenter); both forms of processing have similar effectiveness [64]. Furthermore, the protective effects of inoculation appear to be highly stable over time and may persist for weeks or even months [60, 63, 65]; see, however: [64, 66]. Inoculation treatments are also able to protect attitudes that were not the subject of protection by vaccination but are somewhat related to the target attitude [67, 68]. Additionally, inoculation is effective not only in generating resistance to a change in beliefs but also in shaping and directing them–it can help with developing attitudes in accordance with the desired direction among people who have neutral or negative attitude toward a given issue [69]. Inoculation can cause people to perform the so-called "post-inoculation talk" in which they share with acquaintances both their impressions and thoughts related to the inoculation process and the acquired knowledge while also questioning the opinion of the interlocutors when it is inconsistent with the defended attitude [70, 71]. Therefore, the therapeutic effects of inoculation are not limited to individuals and may contribute to the production of immunity within the communities.

Inoculation is widely used in contexts where a social influence can occur, e.g. in political, health, and social campaigns, public relations, or marketing [55]. Notably, in recent years an effort was made to inoculate people against conspiracy theories [72–74] and misinformation, especially in the context of climate change. For example, Cook et al. [75] used inoculation against misinformation which was introduced in the form of a false balance, where two different positions on a given issue are disproportionately supported (e.g. a scientific consensus of climate scientists vs a single scientist disagreeing with it), and yet are perceived by people as equally credible. The researchers presented the misinformation, but before that participants were presented a text that explained the flawed argumentation technique used by the source of the misinformation (i.e. an explanation of the misinformation strategy used by the tobacco industry) and highlighted the scientific consensus on climate change. The results showed that the inoculation successfully neutralized the negative effects of the misinformation. van der Linden et al. [76], on the other hand, presented the participants a set of statements often used by climate denialists which they were to assess in terms of their familiarity and the degree of being convincing. Previously, some respondents were warned about a potential attack on their attitudes and received a set of charts, graphics, and information confirming the scientific consensus on climate change. The experimenters also discredited some seemingly credible sources of misinformation that can be found in real-world context. This study also managed to reduce the harmful effects of misinformation by using the inoculation technique, which was also more effective than only providing g information about the scientific consensus. Inoculation based on the methods of critical thinking is also effective in counteracting misinformation [77]. Roozenbeek & van der Linden [78] developed a psychological intervention in the form of the online browser game Bad News based on the assumptions of the inoculation theory, in which the player plays the role of a news editor, whose task is to create fake news using techniques used in creating and disseminating misinformation. By creating various types of fake news, players learn about the mechanisms of misinformation and, at the same time, undergo the inoculation process themselves which, consequently, reduces their own vulnerability to misinformation.

Although inoculating against misinformation has been an important and interesting research topic for several years, the previous studies cited above are limited to interventions against misinformation already existing in the public space. Although for many–especially practical–reasons, this seems to be the right endeavor, it seems necessary to check the influence of inoculation on misinformation under laboratory conditions.

In a recent study, Tay et al. [79] used fictional (but based on a real-world topic) misinformation about fair trade. The effectiveness of inoculation located prospectively, i.e. before the misinformation is presented (*prebunking*), and retrospectively, i.e. after the misinformation is presented(*debunking*) was investigated. Although both types of interventions seemed to be effective in reducing CIE, contrary to the researchers' expectations, it turned out that retrospective inoculation was more effective than prospective one (also: [4, 80]; see, however: [74, 81]). In addition to the questionnaire measures, the impact of misinformation on a number of behavioral indicators (here: consumer behavior) was also verified. However, the differences between the conditions were small or statistically insignificant. Though, while this study used fictional material, the procedure was not entirely a "typical" CIE paradigm in a form used in strictly laboratory studies that takes a form of a narrative scenario (e.g. [6, 8, 10]; cf. [82]). Meanwhile, we think that investigating the effects of inoculation on CIE in the "classic" narrative paradigm is useful because the potentially beneficial effects of "vaccination" on neutral materials could broaden the scope of generalizing the results to a variety of contexts and also offer insight into the mechanisms behind CIE.

If inoculation was able to affect fictitious, neutral misinformation, it could mean that it is also capable of modulating memory processes. Since inoculation is able to increase the availability of attitudes [60, 62], thus influencing the resistance to persuasion, similar effects can be expected in counteracting the influence of misinformation. Counterarguments might help to increase the availability of the correction, making it easier to retrieve. This may in turn lead to filter strongly activated but incorrect information more carefully. Alternatively, though not contrary to the previous interpretation, the possible therapeutic effect of inoculation could mean that misinformation processes are not purely due to memory error. While there is no doubt that memory mechanisms are involved in CIE, participants remember the retraction perfectly well in the vast majority of cases. Though it is possible that the other cognitive processes are crucial, that could lead to the creation of something like "memory attitudes", i.e. the relationship of information with its evaluation [83, 84]. Since inoculation affects attitudes, its potential influence on CIE could provide suggestions supporting the idea that CIE is more the result of processes other than memory errors.

Regarding that interpretation, there is one factor that may come in useful. As was mentioned previously, source reliability plays a great role in the context of CIE. Guillory & Geraci [16] verified two aspects of source reliability: trustworthiness and expertise [85]. It turned out that only retractions from the source high in trustworthiness were effective, while the expertise dimension did not influence the effectiveness of corrections. This effect was later replicated in other research [50], even when the expert source was operationalized differently (as being able to make accurate statements based on competence and knowledge resulting from experience and education [19]; in contrast, Guillory & Geraci [16] assumed that the expert source was simply able to access true information). As the information source significantly influences persuasion and belief formation [85–87], the potentially different effects of inoculation on reliance on misinformation when the source reliabili of retractions differs could support the idea of "memory attitudes". Inoculation could prompt the respondents to pay more attention to the sources of corrections which are, sometimes, overlooked in other situations [88]. If non-inoculated participants rely only on trustworthiness, ignoring the expertise dimension, vaccination could make participants be more aware of all source characteristics. Thus, the significant reduction of misinformation reliance and decreased belief in misinformation may be expected if one analyzes if the source is truly credible (i.e. high in both dimensions), but not when the source is only trustworthy or only expert. In the latter case, it could be expected that inoculation would not have any significant impact in reducing misinformation reliance and belief.

This study aims to investigate the impact of inoculation on the processing misinformation and retractions whose sources differ in terms of their reliability. The experiment was based on the method used by Ecker & Antonio [19]. The subjects read the scenario sets, answered open-ended questions, and assessed belief in misinformation and retractions. Half of the participants had previously undergone the inoculation procedure. When it comes to replication hypotheses in misinformation reliance, we expected the CIE to occur (e.g. [6, 10, 17] (Hypothesis 1) as well as the effectiveness of retractions only from trustworthy but not expert source [16, 19, 50] (Hypothesis 2). We also expected different vaccination effects depending on the reliability of the sources of the retractions: the inoculated individual could rely less on misinformation if the source of the correction was credible in both dimensions but not if it was only expert or only trustworthy (Hypothesis 3). This is because inoculation may induce participants to pay more attention to the sources of retractions–in other situations people tend to ignore the sources [88] or tend to be guided only by the trustworthiness dimension [16, 19, 50]. Thus, vaccinated participants can analyze the source credibility and cannot trust retractions from the sources that are not both highly trustworthy and highly expert.

When it comes to belief estimations, we expected to replicate the tendency of believing more in misinformation than in retraction regardless of the reliability of the source [19, 23] (Hypothesis 4). However, as inoculation can shape one's conviction in the direction consistent with protected attitude [69], we expected to observe a reversal of the tendency described above, but only for the highly credible sources (Hypothesis 5). The reasons for this are the same for Hypothesis 3, i.e., that inoculation could prompt to analyze retractions' sources more thoroughly and increase their belief in truly credible information (high in both dimensions). Confirmation of the hypotheses could support the idea that CIE can be a result of rather non-memory factors such as attitudes than memory-like mechanisms.

## Method

The study was approved by the Research Ethics Committee of the Institute of Psychology, Jagiellonian University, Kraków, Poland. Decision no.: KE/29_2021.

### Design

The experiment used a within-between factors design; the within-subjects factors were three types of retraction source reliability: (1) credible source (high in expertise and high in trustworthiness), (2) expert source (high in expertise, but low in trustworthiness), and (3) trustworthy source (high in trustworthiness, but low in expertise); in addition, there were two control conditions: a no-retraction condition and a no-misinformation condition (and therefore also no-correction). The between-subject factor was the inoculation: its presence or absence.

Participants were presented with one scenario for each of the five conditions (for more details see: Materials & procedure), except for the no-misinformation condition which included two scenarios per participant (to equalize the number of scenarios with and without retraction). Participants' reliance on the critical information was measured using an open-ended questionnaire (30 open questions; 5 per scenario) which required inferential reasoning. In addition, six questions that measured participants' belief in the critical information and their belief in the retraction directly (two questions for each scenario that included retraction) were used.

### Power analysis

An *a priori* power analysis was performed to determine the sample size necessary to detect a significant within-between interaction, with α assumed to be 0.05, and desired power: 95%.

Analysis was performed by means of the G*Power software [89], with the option for effect size specification chosen 'as in SPSS'. The analysis was performed for the three commonly assumed effect sizes: small, $\eta^2 = 0.01$ ($f(U) = 0.10$); medium, $\eta^2 = 0.06$ ($f(U) = 0.25$); large, $\eta^2 = 0.14$ ($f(U) = 0.40$). Under these assumptions, the analysis indicated that 464, 76, and 32 participants were required, respectively. Given the available resources, a sample of 141 participants was tested, which allowed for the desired power in the case of large and medium effect sizes, but not for small ones. The power for detecting specific planned comparisons may be somewhat less but still, the sample size was twice as much as required to detect a medium-sized interaction so it should be enough to detect medium-sized planned contrasts.

## Participants

Participants ($N = 141$) were recruited online *via* social media, survey distribution websites, and from Polish university students. The sample consisted of 108 female, 29 male and 4 non-binary participants ($M_{age} = 24.62$; $SD = 7.46$; range 18–68 years). Four people were excluded from the analysis due to a lack of answers in the test, resulting in a final sample of 137 participants (106 female, 28 male, and 3 non-binary; $M_{age} = 24.59$; $SD = 7.56$; range 18–68). Most of the participants were students (70%), 24% had higher education degree and 6% finished only secondary school. No gratification was given for their participation. Written consent was obtained; this method of obtaining consent was approved by the Research Ethics Committee mentioned above.

## Materials & procedure

The study was conducted using Qualtrics (Provo, UT), and was advertised as a survey on the "Memory of narrative texts experiment". The entire experiment took approximately 30 minutes to complete. Participants first read the intentionally misleading description of the research (they were informed that the research was investigating how people process and remember narrations); they then read ethics-approved information, provided consent, and agreed to participate. After completing the information on gender, age, and education they were randomly allocated to one of the 12 possible survey versions (inoculation/lack of inoculation × 6 combinations of scenarios, depending on the layout of conditions).

Six scenarios were constructed based on those previously used in CIE research. Most of them were shortened and modified versions of those used before (water contamination, jewelry theft, warehouse fire, football scandal) and the other two were only inspired by these scenarios (car accident, politician dismissal). The misinformation was introduced indirectly to increase its strength [17]. The content of sentences other than those introducing misinformation and retraction was constructed such that the misinformation could explain the events described. The sources for the three retraction conditions were chosen based on those selected in Ecker & Antonio [19] and Guillory & Geraci [16] research. The expert source was operationalized accordingly to Guillory & Geraci [16] research as a source with access to information (not as a result of professionalism, but simply because of the possibility to access true information); the trustworthy source was operationalized as a source of possible good intentions, but without the access to true information. The credible source was operationalized as a source of high in trustworthiness and high in expertise (usually it was an independent professional). Examples of the sources are presented in the Table 1.

In the no-retraction condition, the target information was introduced without correcting it later; in the no-misinformation condition the target information was never mentioned and there was also no correction. All scenarios existed in both the experimental and the control versions. As in the Ecker & Antonio [19] study, the assignment of scenarios to conditions, and

**Table 1. Examples of the sources (in Football affair scenario\*).**

| Source | Example | Message content |
|--------|---------|-----------------|
| Credible | director of the International Anti-Doping Committee | Three days later, **Olivier Estevez, director of the International Anti-Doping Committee**, announced that Larsson was not involved in the doping affair (. . .) |
| Trustworthy | Popular sports commentator | **Oliver Lindgren, a popular sports commentator**, stated that Larsson was not involved in the doping affair (. . .) |
| Expert | Footballer's manager | **Oliver Lindgren, Larsson's manager**, stated that the player was not involved in the doping affair (. . .) |

*Adapted from Ecker & Antonio [19].

the presentation order of conditions was counterbalanced across participants. Each scenario consisted of 4 fragments–displayed individually on the screen without the possibility of going back to previous ones–containing 1–2 sentences and a total of 30–60 words each.

The main dependent variable was the number of references to target information in the open-ended questions. There were five questions for each scenario. All questions were not based solely on the memory of facts but required the participant to make inferences. The questions were constructed such that the critical information was a possible answer while also allowing for unrelated responses (e.g. "What was the possible cause of the toxic fumes?", for which possible answers would be "The oil paints present in the cabinet", the (mis)information present in the warehouse fire scenario, or an unrelated response such as "Plastic from printers"). The second type of question was rating scales, requiring participants to indicate their level of agreement with a statement on the scale from 0 to 10. These questions concerned only scenarios under retraction conditions and were used to directly measure the participant's belief in the misinformation and in retraction introduced by a specific source (e.g. in a traffic accident scenario where the driver was accused of drunk driving: "To what extent do you believe that the driver was driving under the influence of alcohol?" and "How much do you believe the opinion of the passenger of the car, the driver's colleague, that the driver was not drunk?"). Each participant rated their belief on six scales (two for each of the retraction conditions; one for belief in misinformation and the other for belief in retraction). All questions (both open-ended and rating scales) were presented individually on the screen without the possibility of going back. As an interval task between reading the scenarios and answering the test questions we used filler questions about mood.

Half of the participants also underwent the inoculation procedure, which has been designed in accordance with the theoretical assumptions of the inoculation theory. The procedure consisted of two main parts: (1) Warning and (2) Refutational Preemption. The Warning (specifically a specific warning) was adapted from a study by Ecker et al. [10] and slightly modified. The warning also foreshadowed the presence of an example of misinformation that the participant should be careful about. The second stage, the Refutational Preemption, consisted of two elements. First, a short scenario was presented, that described a fictional debate about banning a potentially harmful substance in food production. However, it was later revealed that various research teams independently confirmed that the substance was in fact completely safe.

After the presentation of the inoculation scenario, there was a question that required the participants to estimate their belief that the described substance should be banned from use (on the 0–10 scale). Depending on the answer, the participant received short feedback. For ratings 0 or 1, participants were informed about their low misinformation compliance. For ratings 2–4, participants were informed about their medium misinformation compliance: message accented both occurrence of some resistance and some acceptance of misinformation.

For higher ratings (5–10), participants were informed about a significant impact of misinformation on them. The second stage of Refutational Preemption followed for all participants in inoculation condition. Examples of misinformation, their debunking, and the mechanisms of reliance on misinformation were presented. Participants were also made aware of the omnipresence of misinformation and the importance of its impact on every aspect of the lives of individuals and societies. Finally, respondents were informed that they would now see several scenarios in which misinformation may or may not occur and that they should answer questions honestly and be careful to not be misinformed.

## Results

### Inoculation question ratings

The average estimate of the belief in the misinformation in the inoculation procedure, i.e. the assessment of the extent to which the described substance is harmful and should be banned in food production, was $M$ = 3.61 ($SD$ = 2.88).

### Questionnaire coding

The open-ended inference questions were coded according to the schema described below. Any unambiguous reference to critical information was scored 1 point (e.g. "The explosions were caused by gas cylinders left in the closet"). The same goes for cases in which the answer could result from the presence of misinformation but misinformation was not mentioned explicitly (e.g. "They should pay attention to the closet in the warehouse" in response to "On what aspect of the fire may the police want to continue the investigation?"). Responses were scored 0 when critical information was not mentioned or explicitly rejected (e.g., "The toxic fumes came from burning plastic." or "The driver may have been drunk, but the allegations turned out to be untrue since no traces of alcohol or drugs were discovered in his blood."). The results for the no-misinformation condition were averaged for the two scenarios per person. The maximum score possible to receive by a participant in one condition was 5 points.

### Inferential reasoning

We performed a repeated measures between-subjects ANOVA, with CIE conditions as within-subject factors and inoculation as between-subjects factor. As the Mauchly's sphericity test showed a significant result ($p < 0.001$, $\hat{\varepsilon} = 0.93$), the Huynh-Feldt correction was used. The analysis showed the interaction effect ($F(3.72, 501.88) = 4.32$, $p = 0.003$, $\eta_p^2 = 0.03$), as well as the main effect of the group ($F(3.72, 501.88) = 58.52$, $p = 0.003$, $\eta_p^2 = 0.30$) and the main effect of inoculation ($F(1,135) = 21.17$, $p < 0.001$, $\eta_p^2 = 0.14$). The mean inference results are presented in Fig 1.

We confirmed the Hypothesis 1 with the replication of the continued influence effect–in the no-inoculation condition, although retraction reduced reliance on misinformation compared to the no-retraction condition regardless of the retraction group ($F$s$(1,135) \geq 3.98$, $p$s $\leq 0.048$, $\eta_p^2$s $\geq 0.03$), it was not enough to eliminate CIE as a significant difference with the no-misinformation condition was observed ($F$s$(1,135) \geq 46.42$, $p$s $< 0.001$, $\eta_p^2$s $\geq 0.26$). However, we failed to confirm the Hypothesis 2 that the effectiveness of retractions would differ depending on their sources–none of them differed significantly from the others ($F$s$(1,135) \leq 3.37$, $p$s $\geq 0.068$, $\eta_p^2$s $\leq 0.02$). This finding contradicts previous reports which stated that trustworthy retraction sources are more effective in reducing CIE than expert sources [16, 19, 50].

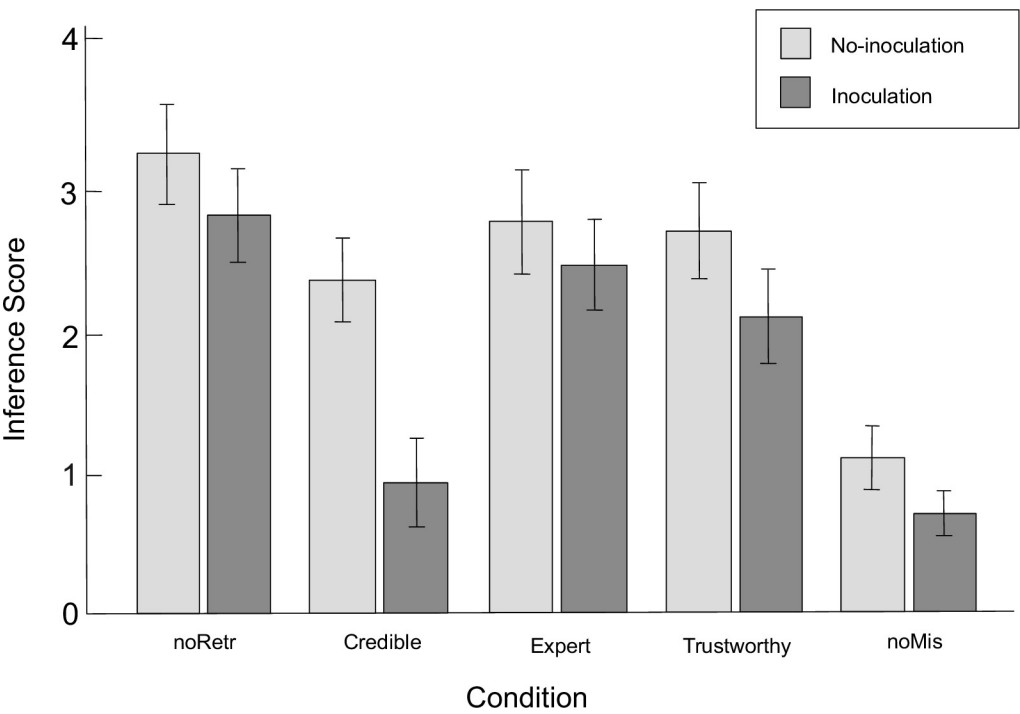

**Fig 1. Inference scores across conditions.** *noRetr* No-retraction condition; *Credible* credible source condition (high in trustworthiness and in expertise); *Expert* expert source condition (high in expertise, low in trustworthiness); *Trustworthy* trustworthy source condition (high in trustworthiness, low in expertise); *noMis* No-misinformation condition. Error bars represent 95% confidence intervals.

As expected, the inoculation affected retraction conditions differently. Hypothesis 3 was confirmed–there was a significant decrease in reliance on misinformation in the credible source condition ($F(1,135) = 34.71$, $p < 0.001$, $\eta_p^2 = 0.21$), as well as some reduction in reliance on misinformation for the trustworthy source condition ($F(1,135) = 4.84$, $p = 0.029$, $\eta_p^2 = 0.04$). However, inference score in trustworthy source condition did not differ significantly from the expert source group ($F(1,135) = 2.89$, $p = 0.092$, $\eta_p^2 = 0.02$). Inoculation did not have a significant effect on the expert source condition ($F(1,135) = 1.29$, $p = 0.259$, $\eta_p^2 < 0.01$). What is particularly important, in the credible source condition, the number of references to misinformation did not differ significantly from the no-misinformation condition ($F(1,135) = 1.63$, $p = 0.204$, $\eta_p^2 = 0.01$), which may indicate the elimination of CIE, although one should be cautious when inferring about lack of differences based on a statistically insignificant result. Moreover, in no-misinformation condition, a significant decrease in the occurrence of responses consistent with critical information was also reported ($F(1,135) = 7.90$, $p = 0.006$, $\eta_p^2 = 0.06$). Overall, while not in all conditions a decline in responses consistent with misinformation could be observed, there was a downward trend in each group, which resulted, for example, in retractions from expert sources being completely ineffective and not differing from the no-retraction condition ($F(1,135) = 2.29$, $p = 0.132$, $\eta_p^2 = 0.02$). In sum, inoculation seemed to be effective in reducing CIE, but only for highly credible sources of retraction.

## Belief ratings

Inoculation was also expected to affect the direct belief in misinformation and in retraction. In accordance to results obtained by Ecker & Antonio [19], in no-inoculation condition, belief in

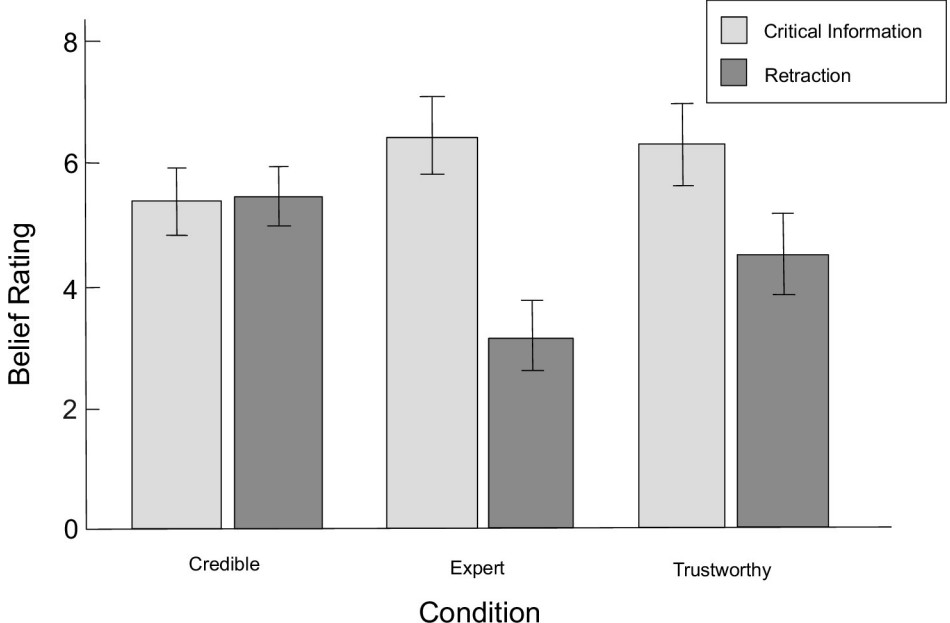

**Fig 2. Mean belief ratings in the no-inoculation condition.** *Credible* credible source condition (high in trustworthiness and in expertise); *Expert* expert source condition (high in expertise, low in trustworthiness); *Trustworthy* trustworthy source condition (high in trustworthiness, low in expertise). Error bars represent 95% confidence intervals.

misinformation was assumed to be higher than belief in retraction, regardless of the reliability of corrections (Hypothesis 4). A two-way ANOVA with repeated measures was performed (within-subject factors: misinformation / retraction × credible source, expert source, trustworthy source; between-subject factor: inoculation / none), in which three-way interaction effect was not showed ($F(2,270) = 2.47$, $p = 0.086$, $\eta_p^2 = 0.02$). Although the main effect of the information was also not found ($F(1,135) = 3.67$, $p < 0.058$, $\eta_p^2 = 0.03$), the main effect of the source was observed ($F(2,270) = 10.40$, $p < 0.001$, $\eta_p^2 = 0.07$), as well as the interaction of information and source ($F(2,270) = 41.75$, $p < 0.001$, $\eta_p^2 = 0.24$), the interaction of inoculation and information ($F(1,135) = 16.65$, $p < 0.001$, $\eta_p^2 = 0.11$), and the interaction of inoculation and source ($F(2,270) = 3.62$, $p = 0.028$, $\eta_p^2 = 0.03$). The mean assessments of direct belief in misinformation and retraction are presented in Figs 2 and 3.

In the no-inoculation condition, as predicted, higher belief in misinformation than in retraction was observed for expert ($F(1,135) = 34.25$, $p < 0.001$, $\eta_p^2 = 0.20$) and trustworthy source condition ($F(1,135) = 10.32$, $p = 0.002$, $\eta_p^2 = 0.07$). Contrary to the expectations, the beliefs were the same for the credible source condition ($F(1,135) = 0.01$, $p = 0.917$, $\eta_p^2 < 0.01$). However, the belief in misinformation was found to be lower and the belief in retraction to be higher for the credible source condition compared to both expert and trustworthy source conditions ($Fs(1,135) \geq 4.48$, $ps \leq 0.036$, $\eta_p^2 s \geq 0.03$). Also, the belief in retraction was higher for the trustworthy than expert source condition ($F(1,135) = 12.96$, $p < 0.001$, $\eta_p^2 = 0.09$), although the belief in misinformation did not differ between these conditions ($F(1,135) = 0.14$, $p = 0.713$, $\eta_p^2 < 0.01$). Therefore, the Hypothesis 4 was confirmed partially.

With regards to Hypothesis 5, a different pattern of results was observed for the inoculation condition. Only for expert source condition more belief in misinformation than in retraction

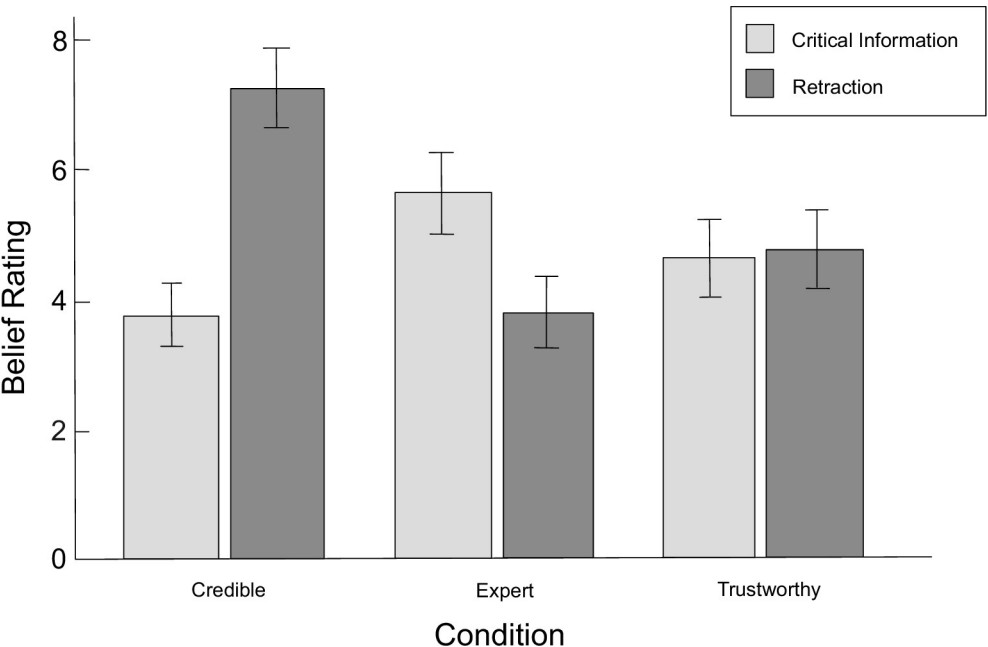

**Fig 3. Mean belief ratings in the inoculation condition.** *Credible* credible source condition (high in trustworthiness and in expertise); *Expert* expert source condition (high in expertise, low in trustworthiness); *Trustworthy* trustworthy source condition (high in trustworthiness, low in expertise). Error bars represent 95% confidence intervals.

was reported ($F(1,135) = 11.10$, $p = 0.001$, $\eta_p^2 = 0.08$), while for trustworthy source condition no differences were observed ($F(1,135) = 0.05$, $p = 0.816$, $\eta_p^2 < 0.01$). For the credible source condition, the belief in retraction was higher than in misinformation ($F(1,135) = 40.90$, $p < 0.001$, $\eta_p^2 = 0.23$). Also, the belief in misinformation was lower, and the belief in retraction was higher for the credible source condition compared to other groups ($Fs(1,135) \geq 4.84$, $ps \leq 0.029$, $\eta_p^2 s \geq 0.04$). Belief in misinformation was lower and belief in retraction was higher for the trustworthy condition than for the expert source condition ($Fs(1,135) \geq 6.56$, $ps \leq 0.012$, $\eta_p^2 s \geq 0.05$). The effect of inoculation on belief in misinformation and retraction, as hypothesized, was observed for the credible source condition, where inoculation significantly increased belief in retraction and lowered belief in misinformation ($Fs(1,135) \geq 12.57$, $ps \leq 0.001$, $\eta_p^2 s \geq 0.09$), which conforms Hypothesis 5. Inoculation also reduced the belief in misinformation for the trustworthy source condition ($F(1,135) = 12.53$, $p = 0.001$, $\eta_p^2 = 0.09$), but did not change the belief in retraction ($F(1,135) = 0.27$, $p = 0.604$, $\eta_p^2 < 0.01$), and also did not seem to affect any of the measures in expert source condition ($Fs(1,135) \leq 2.96$, $ps \geq 0.088$, $\eta_p^2 s \leq 0.02$).

Regression analysis applied to individual sources showed that, under inoculation, belief in misinformation, but not in retraction, significantly predicted reliance on misinformation for the expert source condition ($\beta = 0.32$, $p = 0.028$), similarly to the no-inoculation condition ($\beta = 0.37$, $p = 0.006$). For the inoculation condition in trustworthy source condition, both assumed predictors turned out to be significant (respectively: $\beta = 0.42$, $p < 0.001$ for belief in misinformation, $\beta = -0.26$, $p = 0.025$ for belief in retraction); however, in the no-inoculation condition only belief in misinformation predicted inference score ($\beta = 0.36$, $p = 0.003$). For the credible source condition, in the no-inoculation condition, no relationship between any of the estimates and the inference score was found; however, in the inoculation condition, both

turned out to be significant predictors ($\beta = 0.28$, $p = 0.023$ for belief in misinformation, $\beta = -0.39$, $p = 0.002$ for belief in retraction). Overall, by aggregating the beliefs in misinformation and retraction into two separate variables, it was found that belief in misinformation, but not in retraction, predicted the reliance on misinformation ($\beta = 0.45$, $p < 0.001$, $R^2 = 0.31$) which is the opposite of what was observed by Ecker & Antonio [19]. However, since both estimates showed significant correlations of moderate or high strength with inference scores ($r(135) = 0.54$, $p < 0.001$ for belief in misinformation and $r(135) = -0.39$, $p < 0.001$ for belief in retraction), mediation analysis was performed using the PROCESS v. 3.0 software [90], where the belief in retraction was a predictor, the inference score was the dependent variable and the belief in misinformation was a mediator. The model turned out to be significant: $B = -0.25$, $SE = 0.05$, 95% CI [-0.34, -0.15] for total effect and $B = -0.15$, $SE = 0.04$, 95% CI [-0.23, -0.08] for indirect effect. The direct path between belief in retraction and inference score was not found to be significant, which suggests that belief in retractions negatively influenced relying on misinformation by negatively influencing belief in misinformation.

## Discussion

Contrary to previous research [16, 19, 50], trustworthy sources of retractions did not prove more effective than expert ones in reducing the continued influence effect. However, this is not an unprecedented situation, as Ecker & Antonio [19] in their second experiment did not report any reduction of CIE for trustworthy retraction sources. It is possible that participants were not paying attention to the sources' reliability, which is consistent with the reports of van Boekel et al. [88] who claimed that the reliability of the sources had an impact on processing information only when participants were previously instructed to pay attention to it. This interpretation is also supported by the results of regression analysis which suggested that it is the belief in misinformation, not in retraction, that predicts the inference score. On the other hand, contrary to the results in the expert and trustworthy source conditions, in the credible source condition there were no differences in the estimates of belief in both misinformation and retraction and neither of them was an inference score predictor. It may indicate that the respondents paid some attention to the reliability of the sources.

According to Ecker & Antonio [19] that fact that there was no reduction of misinformation reliance for trustworthy sources of retractions was an effect of increased skepticism about retractions. The skepticism may arise from making belief estimates [91] or from the presence of retraction itself. This is because participants might become suspicious when critical information that perfectly explain the described events is suddenly considered erroneous. This assumption about the causal role of misinformation is present in the mental models theory [6, 25]. Thus, skepticism about the retractions may arise when there is a risk of losing the coherence of the model–as skepticism prevents the creation of a gap while blocking the impact of retractions. The mechanism of this process may be motivational as experiencing discomfort while dealing with retractions may lead to reducing the tension by increasing skepticism about corrections [52]. Also, since the misinformation was introduced implicitly [17], it could increase the tendency to engage in causal thinking, which consequently might neutralize the impact of the reliability of the corrections' sources (see, however [79, 92] for research that failed to replicate Rich & Zaragoza's [17] results).

The main finding of this study is the effectiveness of inoculation in reducing reliance on misinformation in the CIE paradigm. As predicted, the inoculation led to a significant reduction in reliance on misinformation if the retraction was from a highly credible source; it could even lead to the elimination of CIE as the number of references to misinformation in this condition did not differ significantly from the no-misinformation control condition. Inoculation

also caused a slight reduction in reliance on misinformation in the trustworthy source condition. Interestingly, a lower inference score was also observed in the no-misinformation condition. However, as there was no misinformation mentioned, it is difficult to assess why inoculated participants were less likely to refer to absent critical information. It is possible that the inoculation has put participants into a state of heightened general skepticism that has led them to avoid relying on assumptions that might appear plausible and therefore suspicious. In other words, the inoculation procedure may lead people to think that something is "too good to be true". This is in line with the research showing that inoculation generates a general cynicism resulting not only in increased resistance to misinformation but also decreased belief in accurate news, e.g. [93, 94]. On the other hand, if this interpretation was correct, one would also expect a significant decrease in the reliance on critical information in the no-retraction condition, which in our study was not observed. It is possible that the state of increased rejection of critical information is thus observable only when one generates explanation for the events (as was the case in the no-misinformation condition), but not when one is subjected to such an explanation (as was the case in all misinformation conditions). The information being generated may seem easier to reject and has less impact on inferences than external information integral to the story, similar to the results obtained by Johnson & Seifert [24]. However, it is difficult to make unequivocal conclusions.

When it comes to belief estimates, a similar pattern of results to those obtained by Ecker & Antonio [19] was observed. In the no-inoculation condition, the belief in misinformation was significantly higher than the belief in retraction in expert and trustworthy source groups. As noted by Ecker & Antonio [19], greater belief in misinformation may arise from processing retraction in the context of its contradiction with misinformation, which automatically causes it to become less reliable than misinformation. There is also a possibility, as we have argued before, that misinformation is more plausible because it is more consistent with the scenario. Therefore, potential gap creation within the mental model may lead to discomfort, which in turn lowers belief in retraction to prevent gap creation [52]. However, both estimations in the credible source group were found to not differ from each other. Hence, in contrast to the research mentioned, in this paper it is the belief in misinformation that predicted the inference score, not the belief in the retraction. This is in line with the ERP research results obtained by Brydges et al. [95], who concluded that reliance on misinformation may be driven by the strong recollection of the misinformation following poor integration of the retraction into the mental model. Nonetheless, the mediation analysis showed that belief in retraction indirectly influences the reliance on misinformation by lessening the belief in misinformation. As in this experiment the reliability of the retraction sources was manipulated, it seems reasonable that the belief in misinformation should also depend on the level of belief in retraction.

In accordance with our predictions, it was observed that in the credible source condition inoculation reduced belief in misinformation and increased belief in retraction. However, inoculation did not in any way influence the estimations in the expert source condition, at the same time causing the belief in misinformation to lessen in the trustworthy source condition to the level of the belief in retraction. Thus, under the influence of inoculation, it was possible to observe some similarities with previous studies, where the dimension of trustworthiness turned out to be more important than the expertise dimension [16, 19, 50]. Though, there is a possibility that the apparent ineffectiveness of inoculation in the expert source condition may be a result of the critical information not being treated by the participants as misinformation. This is in line with the operational definition of an expert source used in this study, which was defined as one that has access to truthful information but does not necessarily have to have good intentions [16]. The participants could, therefore, for rational reasons, reject the correction and respond in accordance with the critical information [51]. However, this rationality

may be limited to the contextual sequence of the presented scenarios–the inoculation may be also effective for other sources if they precede the credible source. This is because source reliability assessment could result also from its comparison to others. Some confirmation was found for these speculations: in 2 out of 4 cases where the expert and trustworthy conditions appeared before the credible source condition, the inference results in these three groups did not differ from each other ($ps \geq 0.088$). That shows that assessing the reliability of information is not purely analytical, and people are biased towards contextual processing.

## Inoculation against continued influence of misinformation—How does it work?

Interpreting the mechanisms of inoculation on misinformation comes down to analyzing the mechanisms of CIE itself. Both theories described in the introduction offer some promising solutions. According to the mental models account, inoculation could prevent making global inferences from the locally updated model containing the misinformation. However, it seems that there is no reason why one should give up the motivation to maintain consistency in the model and risk creating a gap under the influence of inoculation. Inoculation does not fill the gap because it does not offer alternative solutions [6, 10, 11], but only facilitates the production of resistance against the influence of misinformation. Thus, inoculation may lead to the emergence of a new model (see: [30]). As this is the case only for the credible source condition, it is possible that only in this condition risk of generating a gap existed (maybe because in other conditions retraction couldn't successfully undermine misinformation).

Alternatively, but not mutually exclusive with the previous explanation, according to the selective retrieval account inoculation might work in a similar way to that proposed by Ecker et al. [10] for the pre-exposure warning–actively suppress the automatic activation of misinformation and support the strategic processes in retrieving the retraction thus making it more available [62]. Inoculation could also induce respondents to pay more attention to the sources of corrections, which in other circumstances are overlooked [88]–as it may allow the "antibodies" to correctly recognize the "pathogens" in the critical information when the retraction source is highly credible or in retraction if its source is not reliable (see: [51]).

While both interpretations may reflect possible mechanisms of the influence of inoculation on CIE, we speculated before that the possible therapeutic effect of inoculation could mean that misinformation processes are not purely due to memory-like errors. Since the correction is usually well-remembered, and there are cases where CIE is not expected to occur due to strong belief in the retraction [23], and that in some cases it is rational to rely on misinformation [51], we propose that CIE can be interpreted in line with the memory conversion process.

## CIE mechanisms—A layered account

Hartmut Blank [84] proposed an integrative framework of remembering, describing the stages of *memory conversion* [96] from a memory trace to an observable behavioral manifestation of memory. According to the framework, external factors can influence memory in three different stages. The first stage is accessing **information**, i.e. constructing the memory of an object or event based on traces retrieved from the memory. At this stage, external factors, such as the presence of the specific cues, can make some memory traces more accessible than others, thus resulting in a cue-tuned construction of memory in the form of a representation of the remembered information. Next, at the validation stage, memory information transforms into a **memory belief**, which resembles an attitude or is even identical to it. At this stage, a number of factors may play a crucial role, e.g. informative social influence, attitudes, persuasion, source reliability, activated concepts of Self, etc. Consequently, one may be more likely to accept the

available information if it is in line with one's worldview or corresponds to the opinion of experts or the majority. Finally, at the communication stage, memory belief transforms into a **memory statement**, i.e., a behavioral manifestation of memory (which does not necessarily have to have a verbal form). These statements also may be tuned, mainly because of the normative and socio-motivational influence, e.g. norms of conversation [97], making a person to adapt the way of communicating to the recipient, self-presentation, as well as one's goals and the goals of the remembering itself.

It appears that the remembering framework can be adapted to describing and explaining CIE. Some of its assumptions are consistent with the ongoing theories. The first stage of remembering is consistent with the selective retrieval account, where it is also assumed that misinformation can be retrieved automatically, e.g. due to the presence of appropriate cues [10, 12, 17]. At the same time, it is also assumed that the retrieval stage plays a key role in the occurrence of behaviorally observable misinformation effects. However, adding a validation stage could be helpful in further explanation of misinformation reliance. According to the selective retrieval account, retrieving the retraction helps the monitoring processes to validate misinformation and confirm its falsity [10]. If this process is to be successful, then CIE should not happen. However, if one fails to retrieve the correction, then misinformation monitoring would become impossible and would lead to relying on misinformation. In most cases, however, people are aware of the presence of both misinformation and its retraction yet make use of the misinformation. It should therefore be expected that, after misinformation retrieval, CIE occurs not because the misinformation has not been sufficiently suppressed due to the failure of retraction retrieval, but as a result of the emergence of a memory belief.

The internal representation of the memory task (a certain set of assumptions about the nature and extent of the memorized content, for example, the *consistency assumption* [98].) may play a key role in this process. If one experiences two conflicting pieces of information, one may want to explain the contradiction in order to keep one's assumptions–in this case, for example, by doubting one piece of information and reporting the other. Motivational factors may play a key role in this conflict resolution, e.g. when dealing with the contradiction between misinformation and its correction; see: [52]. Apart from the mentioned *consistency assumption*, one may have a number of different assumptions, such as the *relevance assumption*, according to which some information in the scenario must correctly explain the questions asked (or, more universally, that "there must be some explanation for certain events"; see: [99] to illustrate this mechanism in non-laboratory conditions). Therefore, misinformation is used because it seems to be the most appropriate answer to the problem. However, when these assumptions are overturned, e.g. as a result of a warning or inoculation, the emergence of a different memory belief than if the assumptions had not been disproved might occur [98]. The same may be true for the sources' reliability–if the relevance assumption is not overturned, misinformation may be the best possible answer regardless of the degree of reliability of the source, or the belief in the retraction itself. The idea of the relevance assumption or the coherence assumption also coincides with the general idea of mental models since in both cases causal reasoning plays an important role.

Adjustments made at the communication stage seem to be equally important. Factors that influence the choice of the form of the memory statement can modify the memory report to be even different from the memory belief one possesses [84]. Thus, one can behave in accordance with misinformation even though one declares that one does not believe in this misinformation; contrariwise, belief in misinformation may not show up in behavior. For example, it can be assumed that when people subjected to the CIE procedure answering questions, they are involved in some *problem-solving process* [98] in which they must decide whether their memory beliefs are suitable for use in the report. For a variety of reasons, ranging from the strength

and valence of a memory belief to communicative motivations, they may or may not use them. To the outside observer, CIE occurs or not regardless of whether the memory statement differs from the memory belief or not. Inoculation can also play an important role at this stage–it can influence e.g. decisions to use misinformation in verbal or non-verbal behavior.

As Blank [84] points out, the remembering framework offers the opportunity to understand memory in both cognitive and social terms within a single process. We believe this model is also well-suited to explain CIE in both real-life situations and when misinformation is exclusively fictional. Its frameworks do not deprive the mechanisms of the selective retrieval account of their significance but place them in a broader context and offer new interpretative possibilities, giving an important role to deliberative and motivational factors and not limiting CIE to mechanistic, purely cognitive or memory-like processes, as it is usually conceptualized (cf. [100]). Also, the idea of the need to maintain consistency, without referring to mental models, can be included in relevance and consistency assumptions, according to which one prefers misinformation because it is better suited to the performance of the memory goal of answering questions. What is also important is that the remembering framework also captures situations in which continued reliance on misinformation may be considered rational [51]. Understanding CIE as a layered process may also help to address the relationship between patterns of misinformation compliance at cognitive and behavioral levels, which, as pointed out by Tay et al. [79], may have weak connections. As attitudes and beliefs may weakly and indirectly predict behavior [101], a similar approach to CIE in the remembering interpretation can be attempted.

We propose that the interpretation of CIE should begin with an analysis of the behavioral manifestations of memory as the belief itself is not directly observable. Following Blank [84] we argue that only after excluding factors influencing memory conversion we obtain the possibility of interpreting CIE as the effect of retrieval processes. Otherwise, speculating about CIE mechanisms seems unjustified because, as Blank [84, 98, 102] notes, conversion effects, if unrecognized, can be mistaken for "conventional" retrieval effects and can mislead theoretical conclusions. Of course, this does not mean that CIE does not occur under certain conditions as a result of a retrieval error–rather, we argue that, given the full remembering process, it is simply less likely.

## Practical implications, remarks and future directions

Although we have focused largely on theoretical explanations, this study provides also some clear practical implications. Thanks to inoculation, under conditions where a retraction came from a highly credible source, it was possible to eliminate the influence of misinformation to the level of the control no-misinformation condition. This widens the boundary conditions of the effectiveness of vaccination against misinformation, which may find application in anti-misinformation immunization procedures. Our research may be the first that has used inoculation procedure on CIE in the "classic" narrative paradigm and one of the first that has used fictional misinformation (see: [79]). Thus, there is a possibility that our results can be generalized to a wider range of conditions (e.g., vaccination or climate change). As inoculation highlights misleading argumentation techniques such as selective use of data or use of fake experts (see: [103]), thus providing protection from misinformation, it could be beneficial to stress retraction sources characteristics. Additionally, it would also pay off to boost the trustworthiness of the source, if it is perceived as an expert, but not necessarily trustworthy; that potentially would enhance the inoculation effectiveness. It seems that highlighting the fake experts technique works in a similar way by reducing the trustworthiness of the source (without necessarily having to lower the perceived expertise), so increasing the trustworthiness could also be

beneficial. Finally, the use of the exercise similar to that used in this study (which made the participants aware that they are compliant to misinformation or not)–though not pivotal–may be useful, as in some cases inoculating against misinformation without such exercise seems to be ineffective (cf. [104]).

To fully investigate these therapeutic effects of inoculation, however, a series of further studies would need to be done. The major concern about this study is that we did not conduct any pilot study prior to running the main experiment in order to select retraction sources. Instead, we chose sources partly on the basis of the previous research (e.g. [16, 19]), which may cause some differences between the scenarios, affecting the results. It would be valuable to replicate the results with a previously conducted pilot study. Another issue is that while it was possible to demonstrate a consistent inoculation effectiveness for the highly credible retraction source, a considerable effect was also observed for the remaining retraction conditions if they were presented first before one became acquainted with the most credible source. To investigate this problem in more detail, future studies would need to focus on the extent to which the size of CIE may depend on contextual factors, as well as perform full inoculation using critical and analytical thinking methods [77]. This could be especially beneficial because analytical thinking is positively correlated with the ability to discern fake news from real news [105]. The effectiveness of inoculation could also be compared to only instructing participants not to use retracted content in answering questions, as it is certainly wrong, no matter what source it comes from, and to investigate the effectiveness of inoculation on expert sources other than defined in this study [19]. The usefulness of inoculation should also be verified when the source of the retraction is not specified. By fully controlling the issue of reliability of the sources, as well as examining participants in a between-subject design experiment, it is possible to fully assess to what extent inoculation is able to counteract the impact of misinformation.

Finally, the power of the current sample only allowed to detect medium-sized effects. This must be borne in mind when interpreting the lack of significance of some effects.

## Conclusion

We conducted a study exploring inoculation against the continued influence of misinformation. Results have shown that inoculation is a highly effective technique to reduce the impact of misinformation reliance when the sources of retraction are highly credible. In order to interpret both our results and the CIE mechanisms themselves, we proposed a theoretical framework of remembering [84], which describes the conversion of memory from a memory trace to a behavioral statement. Both the impact of inoculation on misinformation reliance observed, as well as the proposed theoretical approach could make a significant contribution to the attempts to further understand the mechanisms of misinformation and ways of immunizing against it.

## Supporting information

**S1 Data. Raw data.**
(XLSX)

**S1 File. Materials used in the study.**
(DOCX)

## Author Contributions

**Conceptualization:** Klara Austeja Buczel.

**Data curation:** Klara Austeja Buczel.

**Formal analysis:** Klara Austeja Buczel.

**Investigation:** Klara Austeja Buczel, Paulina D. Szyszka, Adam Siwiak.

**Methodology:** Klara Austeja Buczel, Adam Siwiak.

**Project administration:** Klara Austeja Buczel.

**Software:** Klara Austeja Buczel, Paulina D. Szyszka.

**Supervision:** Malwina Szpitalak, Romuald Polczyk.

**Validation:** Klara Austeja Buczel, Adam Siwiak, Malwina Szpitalak, Romuald Polczyk.

**Visualization:** Paulina D. Szyszka.

**Writing – original draft:** Klara Austeja Buczel, Paulina D. Szyszka, Adam Siwiak, Malwina Szpitalak.

**Writing – review & editing:** Klara Austeja Buczel, Romuald Polczyk.

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
