## [Decision Letter · Decision Letter 0]

23 Aug 2021

PONE-D-21-17666

Vaccination against misinformation: The inoculation technique reduces the continued influence effect

PLOS ONE

Dear Dr. Polczyk,

Thank you for submitting your manuscript to PLOS ONE. After careful consideration, we feel that it has merit but does not fully meet PLOS ONE’s publication criteria as it currently stands. Therefore, we invite you to submit a revised version of the manuscript that addresses the points raised during the review process.

ACADEMIC EDITOR:

As you can see below, both expert reviewers found your study interesting and offer extensive and extremely valuable comments to improve your manuscript. However, while both have identified some merits, there are also conceptual and methodological issues that should be fully addressed. Overall, from my own assessment, I agree with most of the presented comments. I am not going to reiterate them all. Still, but I would suggest particular attention to the following:

Moderate your claims and do not overstate your findings

Stick to the terms/concepts (sometimes redundancy is a good thing)

Provide details on the Power calculation, namely the effect size specification

Please consider the additional references suggested by the reviewers

All the remaining comments of the reviewers should be comprehensively addressed.

We look forward to receiving your revised manuscript.

Kind regards,

Margarida Vaz Garrido

Academic Editor

PLOS ONE

Journal Requirements:

Reviewers' comments:

Reviewer's Responses to Questions

**Comments to the Author**

1. Is the manuscript technically sound, and do the data support the conclusions?

Reviewer #1: Partly

Reviewer #2: Yes

2. Has the statistical analysis been performed appropriately and rigorously? 

Reviewer #1: Yes

Reviewer #2: Yes

3. Have the authors made all data underlying the findings in their manuscript fully available?

Reviewer #1: Yes

Reviewer #2: Yes

4. Is the manuscript presented in an intelligible fashion and written in standard English?

Reviewer #1: No

Reviewer #2: Yes

5. Review Comments to the Author

Reviewer #1: Review of PONE-D-21-17666

1. Abstract: Rephrase “new highly applicable technique has been discovered” – You hardly discovered this technique.

2. p.3: Any reference to Nyhan & Reifler’s 2010 study must be accompanied by a reference to a failure to replicate their work, e.g. https://doi.org/10.1007/s11109-018-9443-y.

3. p.3 reference to the integration accounts of the CIE should refer to Kendeou’s KReC model, e.g. https://psycnet.apa.org/record/2014-41945-016 and https://doi.org/10.3758/s13421-018-0848-y.

4. p.4 “it is assumed that the model updating leads to local consistency and to errors in drawing conclusions when the situation is assessed globally” requires more explanation.

5. p.4: The selective retrieval account does not *necessarily* require the assumption of dual processes.

6. p.4: “CIE occurs when both misinformation and its correction are activated” – a CIE could also occur when the correction is not activated at all.

7. p.4/5: “controlled processes based on cognitive resources” is an odd phrase. Of course cognitive processes are based on cognitive resources.

8. p.5: illusory truth means familiar (e.g. repeated) information is more likely perceived as true, not “information known beforehand”

9. p.5: When discussing the potential role of worldview effects, the discussion needs to be more balanced, as there is also evidence that worldview does not matter much. For a recent discussion, see https://doi.org/10.1098/rstb.2020.0145

10. p.5/6: When discussing factors that are useful to reduce misinformation reliance, consider consulting https://doi.org/10.1371/journal.pone.0210746.

11. p.6 It is incorrect to state that “no novel techniques limiting the impact of misinformation in the CIE paradigm were identified”, not only in light of my previous comment, but also in light of existing research on inoculation in this domain. Do not overstate the novelty of your work please.

12. p.8: it should be mentioned that in Cook et al.’s study, the inoculation led to generalization from tobacco to climate change.

13. p.9: when claiming that no research has tested inoculation against neutral/fictional misinformation, it may be worth discussing 10.31234/osf.io/48zqn, which used fictional misinformation (albeit about a real-world topic). This is also relevant as it failed to replicate Rich & Zaragoza’s implied vs. explicit misinformation difference referred to on p.12.

14. p.11: I cannot replicate the power analysis. To detect a 5*2 within-between interaction effect of f = 0.1 with alpha = 0.05 and beta = 0.15 requires a sample of N = 340, according to my G*Power analysis. Note that even greater power is needed to test some of the specific hypotheses, so technically I would argue the study is underpowered, especially given how much emphasis is placed in the discussion on some of the simple contrasts between two specific cells (and given no correction for multiple comparisons is applied). If an (online) replication is possible, I would strongly suggest that, even if it were just one that focuses on key conditions. If not, achieved power should be discussed as a limitation.

15. p.13 I was initially confused about the “six rating scales” – there seem to be two scales per scenario (one targeting the misinformation, one the retraction), so with six scenarios, I thought there should be 12 scales, but there are only 3 retraction scenarios, so it’s 2*3. Please add some additional clarification.

16. p.15 (but also throughout): All examples should make sense to a reader not familiar with the materials. This could be achieved by sticking to one scenario across examples. For example, it is unclear what role “Harter's son” plays in the theft scenario.

17. p.15 an ANOVA with two factors is not a one-way ANOVA.

18. p.15: There is no “main effect of the interaction” – it’s an interaction effect. On p.17, is “main interaction effect” meant to mean “three-way interaction effect”?

19. Figures: Apart from the obvious issue with the y-axis labels, I think the no-correction control condition should be presented all the way on the left; also the error bars need to be specified.

20. p.17 please rephrase, avoiding the exaggerated language (“proved”, “extremely”): “inoculation has proved to be extremely effective in reducing CIE, but only for highly credible sources of retraction”. Again, a replication would be required to make any strong claims. Also, “prove” on p.21 is too strong.

21. p.19: The finding that “belief in misinformation, but not in retraction, predicted the reliance on misinformation” meshes well with the ERP results here: https://doi.org/10.1080/20445911.2020.1849226

22. p.21 : I did not understand how “Skepticism could be the result of maintaining the coherence of the model, which prevents the creation of a gap while blocking the impact of retractions.”. Please unpack this for the reader.

23. p.22 I agree with the conclusion that it seems that an inoculation “put participants into a state of

heightened general skepticism” – consider cross-referencing this with other literature on response criterion effects. Did inoculated participants generally write fewer words?

24. p.23 I did not understand why it could be considered “right to rely on critical information” in the expert-correction condition?

25. p.23 I did not follow the discussion on generated information, because participants in this study did not generate the information. I did not understand why information generation was (related to?) “the reason why participants rejected the critical information which "came to their minds" more easily than one that was just a part of a story, which just was not interpreted as a "candidate for being false"”. Please unpack and rephrase.

26. p.24 The notion that repeating misinformation in the context of a correction may lead to increased misinformation reliance is contradicted by a number of studies (e.g., https://doi.org/10.1016/j.jarmac.2017.01.014, https://doi.org/10.1186/s41235-020-00241-6).

27. p.25 It is unclear what “inoculation does not immunize to the scenario schema” means.

28. It may be good to add a note to either introduction or discussion that the two theoretical CIE accounts are not mutually exclusive and may be complementary.

29. The discussion is much too long, and could be easily cut in half. It contains too much redundant summary, and tends to place too much weight on specific condition differences that the study was not powered to assess properly.

30. While the English is largely acceptable, there were a few sentences that were confusing.

a. Abstract: This sentence is confusing: “The results showed that the reliability of the sources of corrections did not affect their processing when participants were not inoculated, but, at the same time, a significant reduction in the reliance on misinformation among vaccinated participants when the correction was made from a highly credible source was observed.” Break down into 2 sentences, starting the second with “When participants were inoculated…”

b. Abstract: What does “within the remembering framework” mean? Rephrase.

c. p.4: “where the whole model is reconstructed and new is created” should read “where the whole model is reconstructed and a new one created” or “where the whole model is reconstructed and created anew”

d. p.5: “information being coded is always treated as it was true and may be falsified later by attaching a negation tag to itself” should read “information being encoded is always treated initially as if it were true and is only falsified subsequently by attaching a negation tag” (information cannot attach anything to itself)

e. p.5: “non-memory factors, e.g. motivational (e.g. attitudes and the worldview)” should read “non-memory factors such as motivational factors (e.g., a person’s attitudes and worldview)”

f. p.7: “attitudes that are not vaccinated” – attitudes cannot be vaccinated

g. p.10: This is confusing—consider rephrasing and splitting into two sentences: “Given that retractions are more effective if their source is trusted but not necessarily expert, not only replication of existing reports can be expected (Hypothesis 1), but also different vaccination effects depending on the reliability of the sources of the retractions.”

h. p.14: This needs revision: “For ratings 0 or 1, message informing of surviving the misinformation attack was presented; for ratings 2-4 it informed about the occurrence of resistance, but the impact of misinformation on the participant’s decision was accented”

i. p.19: I think “The effect of inoculation on belief in misinformation and retraction, as hypothesized, was observed for the credible source condition, whereas inoculation significantly increased belief in retraction and lowered belief in misinformation” needs to read “The effect of inoculation on belief in misinformation and retraction, as hypothesized, was observed for the credible source condition, *where* inoculation significantly increased belief in retraction and lowered belief in misinformation”

31. The discussion section was particularly difficult to follow. Consider the following section as an example: “On the other hand, the results for credible source condition, in which there were no differences in the estimates of belief and in which neither of these variables was an inferential reasoning score predictor, may indicate that the respondents paid some attention to the reliability of the sources. Therefore, skepticism towards corrections may play a greater role considering that even with lower estimates of belief in misinformation compared to the other two retraction conditions, the results of inference in the credible source condition did not differ from them. As a result, one may express the same estimates of belief, but skepticism requires them to be careful about the submitted memory reports and provide the answer that best explains the described events. The skepticism itself may result from the

fact that a situation where the critical information perfectly explaining the described events may be considered, for some reason, erroneous, is thought to be suspicious by participants.” – I don’t really understand any of the four sentences in that section. Simplify the sentence structures. Shorten the sentences where possible. Avoid referring to “these variables” or “them”, instead always specify what you mean. Choose one set of terms and then stick to those terms; e.g. stick to the term “inference score” consistently, without paraphrasing (inferential reasoning score); what are “results of inference”? Also shorten the paragraphs, as there are multiple paragraphs that span more than a page. Another example is on p.24: What does “its” refer to in “Retraction may therefore be less credible as it would lead to its creation”? I stress these are merely examples; the entire discussion requires the authors’ careful attention.

32. There are many typos and minor language issues throughout, but these can probably be dealt with by the editorial office. Just pointing out a typo that might escape them on p.15: Huyhn

Reviewer #2: Study tests the impact of inoculation on reducing the continued influence effect (CIE) – finding that when participants are inoculated before showing misinformation and a retraction, it reduces reliance on misinformation to the same level as a control group that weren’t exposed to the misinformation (e.g., inoculation combined with a retraction potentially eliminates the CIE). This is an interesting and insightful result, in a well-designed experiment, and worthy of publication. The connection of inoculation theory with CIE is novel.

I have no problems with the experiment or results. Generally, there is one omission in the discussion/conclusion that I would like to see addressed. The inoculation seems to warn against the general possibility of being misinformed, which is not an optimal way to inoculate people against misinformation as it can breed general cynicism resulting in not only decreased vulnerability to misinformation but also decreased belief in accurate news. There is a body of literature exploring the breeding of cynicism when interventions against misinformation potentially decrease trust in accurate news sources that is relevant to this discussion (Ashley, Poepsel, & Willis, 2010; Mihailidis, 2009; Pennycook & Rand,2017; Tully & Vraga, 2017; Van Duyn & Collier, 2017).

6. PLOS authors have the option to publish the peer review history of their article (what does this mean?). If published, this will include your full peer review and any attached files.

Reviewer #1: No

Reviewer #2: No

---

## [Author Response · Author response to Decision Letter 0]

6 Nov 2021

RE: Revised version of the article: “Vaccination against misinformation: The inoculation technique reduces the continued influence effect”

Dear Editors,

Please find enclosed the revised version of our article. Below we listed all changes that were made:

Major changes:

1. We shortened the discussion part by about 1/3 of its previous length and made it clearer.

2. We corrected some linguistic mistakes thorough the text.

3. Aside from the research mentioned by reviewers, we added in the introduction and in the discussion sections some new research references:

a. https://doi.org/10.3758/s13421-021-01232-8

b. https://doi.org/10.3758/s13421-011-0179-8

c. https://doi.org/10.1073/pnas.2020043118, https://doi.org/10.1111/jcom.12171

d. https://doi.org/10.1080/01296612.2017.1384145

e. https://doi.org/10.1080/17437199.2010.521684

f. Connor Desai SA. (Dis) continuing the continued influence effect of misinformation [Doctoral thesis]. [London]: City, University of London; 2018.

g. Connor Desai SA, Reimers S. Some misinformation is more easily countered: An experiment on the continued influence effect. In: Proceedings of the 40th Annual Meeting of the Cognitive Science Society. Austin, TX: Cognitive Science Society; 2018. p. 1542–7.

We also tried to attend to all comments and recommendations made by the reviewers. Here is the list of the specific changes and corrections:

Reviewer A:

1. We corrected the fragment “new highly applicable technique has been discovered” to “certain boundary conditions for inoculation efficiency have been discovered” (p. 2).

2. We added a reference to the research mentioned and also https://doi.org/10.1186/s41235-020-00241-6 when it comes to failure to replicate Nyhan & Reifler’s (2010) study (p. 3).

3. We added references mentioned about the KReC model to the discussion of the integration accounts (p. 4).

4. We rephrased the sentence „it is assumed that the model updating leads to local consistency and to errors in drawing conclusions when the situation is assessed globally” into „In the case of CIE, it is assumed that the model updating leads to local consistency but conclusions are drawn globally, leading to errors due to local sustaining of misinformation” (p. 4).

5. We rephrase some phrases in the description of the selective retrieval account to make it clear that this approach does not necessarily require the assumption of dual processes and that the CIE could also occur when the correction is not activated (p. 4 / p. 5).

6. We have deleted the fragment „based on cognitive resources” from “controlled processes based on cognitive resources” (p. 5).

7. We corrected the description of the illusory truth effect (p.5)

8. Aside from mentioned https://doi.org/10.1098/rstb.2020.0145, we added also https://doi.org/10.1098/rsos.180593, https://doi.org/10.1098/rsos.160802, https://doi.org/10.1111/pops.12586, when it comes to discussing the role of worldview effects to make the discussion more balanced (p. 5).

9. We mentioned https://doi.org/10.1371/journal.pone.0210746 to the discussion of factors useful to reduce misinformation reliance (p. 6).

10. We change some phrases about the fragment where techniques of limiting the CIE were discussed, that previously claimed that „no novel techniques limiting the impact of misinformation in the CIE paradigm were identified”. We meant the particular laboratory „narrative paradigm” of CIE, as it is usually performed (Ecker et al., 2010; Johnson and Seifert, 1994) and as it was defined by Connor Desai (2018) in her work. Nevertheless, we added also some clarification about it when discussing Tay et al.’s (2021) study (p. 9).

11. We mentioned that in the Cook et al.’s (2017) study the inoculation led to the generalization of resistance from the explanation of strategy used by the tobacco industry to misinformation about climate change (p. 8).

12. We discussed Tay et al.’s (2021) 10.31234/osf.io/48zqn work about the inoculation against fictional misinformation (p. 9). We also refer to this study in the discussion when it comes to failing to replicate Rich & Zaragoza’s implied vs. explicit misinformation difference (p. 22) and in the discussion about the link between misinformation compliance at cognitive and behavioral levels (p. 29).

13. We repeated the power analysis in the G*Power, but we achieved different results than You did. Perhaps it happened because You changed the content of the field „Number of groups” to the field „Number of measurements”. Note that in our study groups were 2 and measurements were 5. We attach a screenshot of our calculations below. 

14. We added clarification about the rating scales. The rating scales concerned only the 3 retraction conditions. Each participant had access to only one scenario for each retraction condition, so the scales were for only three scenarios. As there were 2 scales for each scenario, 6 scales per person are obtained (p. 14).

15. We added changes when it comes to the examples of the participants’ responses – now we stick only to two scenarios (warehouse fire and car accident) (p. 14 and throughout).

16. We deleted the „one-way” from the ANOVA description (p. 16).

17. We rephrase „main effect of the interaction” to „interaction effect” (p. 16).

18. We changed Figure 1 that the no-correction control condition is now presented on the left (we also made appropriate changes to the description of the figure) and specified error bars in the descriptions of figures. We also found out that the issues with y-axis labels were due to the conversion of the files from the .cdr to .eps. We couldn't fix this in the EPS file, so we're sending the corrected figures in a PDF version.

19. We corrected errors resulting from the translation from Polish to English, e.g. exaggerated language (throughout the text).

20. We added a reference to the Brydges et al.’s (2020) https://doi.org/10.1080/20445911.2020.1849226 study when it comes to the finding that belief in misinformation predicted the reliance on misinformation (p. 23).

21. We rephrased and added clarification when it comes to speculations about skepticism as a result of maintaining the coherence of the mental model, also adding a reference to the motivational factors (https://doi.org/10.3758/s13421-021-01232-8) (p. 21).

22. We did our best but unfortunately we couldn’t find any research concerning both response criterion effects and skepticism.

23. In the response to the comment about why „it could be considered “right to rely on critical information” in the expert-correction condition?”: we have deleted the fragment (p. 22) while shortening the discussion part, but in the latter text (p. 24) there is an explanation of that reasoning with the reference to the Connor Desai et al.’s (2020) study, in which authors concluded that under certain conditions (e.g. when the source of the retraction is unreliable) it may be considered rational to rely on misinformation. In this study, the expert source of the retraction was operationalized as in Guillory & Geraci’s (2013) study, and therefore it could be considered by participants as unreliable - thus it may be rational to rely on critical information.

24. We rephrased the part about generated information to be more clear (p. 22).

25. We deleted the fragment about the repetition of misinformation in the context of a correction that may lead to increased misinformation reliance (previously on p. 24).

26. We deleted the fragment „inoculation does not immunize to the scenario schema” as it was not clear and redundant, and also in the process of shortening the discussion section (previously on p. 24).

27. We added a note that two theoretical CIE accounts are not exclusive and may be complementary both in the introduction (p. 3) and in the discussion (p. 25).

28. We corrected some sentences mentioned:

a. “The results showed that the reliability of the sources of corrections did not affect their processing when participants were not inoculated, but, at the same time, a significant reduction in the reliance on misinformation among vaccinated participants when the correction was made from a highly credible source was observed.” was broke down into: „The results showed that the reliability of the sources of corrections did not affect their processing when participants were not inoculated. However, under inoculation condition, a significant reduction in the reliance on misinformation among vaccinated participants when the correction was made from a highly credible source was observed.” (p. 2).

b. We added clarification what “within the remembering framework” means by adding to it sentence: “within the remembering framework describing the conversion from memory traces to behavioral memory statements” (p. 2).

c. The sentence “where the whole model is reconstructed and new is created” was changed to „where the whole model is reconstructed and a new one is created” (p. 4).

d. The sentence “information being coded is always treated as it was true and may be falsified later by attaching a negation tag to itself” was changed into “information being encoded is always treated initially as if it were true and is only falsified subsequently by attaching a negation tag” (p. 5).

e. The sentence “non-memory factors, e.g. motivational (e.g. attitudes and the worldview)” was changed into “non-memory factors such as motivational factors (e.g., person’s attitudes and worldview)” (p. 5).

f. We rephrased sentence containing “attitudes that are not vaccinated” to the „Inoculation treatments are also able to protect attitudes that were not the subject of protection by vaccination but are somewhat related to the target attitude” (p. 7).

g. The sentence “Given that retractions are more effective if their source is trusted but not necessarily expert, not only replication of existing reports can be expected (Hypothesis 1), but also different vaccination effects depending on the reliability of the sources of the retractions.” was changed into „Given that retractions are more effective if their source is trusted but not necessarily expert (16,19,50), replication of existing reports can be expected (Hypothesis 1). We also expected different vaccination effects depending on the reliability of the sources of the retractions.” (p. 10).

h. The fragment “For ratings 0 or 1, message informing of surviving the misinformation attack was presented; for ratings 2-4 it informed about the occurrence of resistance, but the impact of misinformation on the participant’s decision was accented” was rephrased into „For ratings 0 or 1 message informed participants about their low misinformation compliance. For ratings 2-4 message informed participants about their medium misinformation compliance: it accented to both occurrence of some resistance and some acceptance of misinformation” (p. 15).

i. We corrected a mistake in the sentence “The effect of inoculation on belief in misinformation and retraction, as hypothesized, was observed for the credible source condition, *whereas* inoculation significantly increased belief in retraction and lowered belief in misinformation” (p. 19).

29. As possible, due to comments, we tried to simplify sentence structures, specified variables when referring to them, and unified terms used in the discussion section.

30. We corrected a typo „Huyhn” into „Huynh” (p. 16).

Reviewer B

1. We added reference to the research on cynicism and decreasing trust in the accurate news as a result of the inoculation procedure (https://doi.org/10.1080/15205436.2018.1511807, https://doi.org/10.1080/08838151.2016.1273923) in the discussion section (p. 22).

Yours sincerely

Romuald Polczyk

---

## [Decision Letter · Decision Letter 1]

10 Jan 2022

PONE-D-21-17666R1Vaccination against misinformation: The inoculation technique reduces the continued influence effectPLOS ONE

Dear Dr. Polczyk

Thank you for submitting your manuscript to PLOS ONE. After careful consideration, we feel that it has merit but does not fully meet PLOS ONE’s publication criteria as it currently stands. Therefore, we invite you to submit a revised version of the manuscript that addresses the points raised during the review process.

I commend your efforts in revising the manuscript, which is now greatly improved. Below you can find the reviewers’ comments. As you can see, these are mostly minor points that you should be easily able to address. However, I emphasize the importance of addressing all the comments provided, particularly the power issues (following the recommendations of Rev 1) and the discussion of source characteristics and clarification of the hypotheses (suggested by Rev 2).

Please consider the additional references suggested by the reviewers

Please proofread the paper (Rev 1 offers several suggestions in this regard)

We look forward to receiving your revised manuscript.

Kind regards,

Margarida Vaz Garrido

Academic Editor

PLOS ONE

Journal Requirements:

Reviewers' comments:

Reviewer's Responses to Questions

**Comments to the Author**

1. If the authors have adequately addressed your comments raised in a previous round of review and you feel that this manuscript is now acceptable for publication, you may indicate that here to bypass the “Comments to the Author” section, enter your conflict of interest statement in the “Confidential to Editor” section, and submit your "Accept" recommendation.

Reviewer #1: (No Response)

Reviewer #3: (No Response)

2. Is the manuscript technically sound, and do the data support the conclusions?

Reviewer #1: (No Response)

Reviewer #3: Partly

3. Has the statistical analysis been performed appropriately and rigorously? 

Reviewer #1: (No Response)

Reviewer #3: Yes

4. Have the authors made all data underlying the findings in their manuscript fully available?

Reviewer #1: (No Response)

Reviewer #3: Yes

5. Is the manuscript presented in an intelligible fashion and written in standard English?

Reviewer #1: (No Response)

Reviewer #3: Yes

6. Review Comments to the Author

Reviewer #1: Review of PONE-D-21-17666-R1

The authors should be commended for a thorough revision. I only have a few remaining points, all minor.

1. I believe the power analysis was conducted incorrectly. Effect sizes provided by commercial statistics programs already account for correlation among repeated measures. To avoid this, the option “as in SPSS” should be selected. This returns N = 340. If the authors believe this is incorrect, they need to at least report and justify their chosen value for the correlation. Note that even greater power is needed to test some of the specific hypotheses, so technically I would argue the study is underpowered, especially given how much emphasis is placed in the discussion on some of the simple contrasts between two specific cells (and given no correction for multiple comparisons is applied). If an (online) replication is possible, I would strongly suggest that, even if it were just one that focuses on key conditions. If not, achieved power should be discussed as a limitation.

2. Abstract: Unclear what “behavioral memory statements” are—perhaps use “behavioral manifestations of memory” as in the GD?

3. P.13: fix odd format of “Guillory & Geraci”

4. P.21 “most likely be an effect” should be “most likely an effect”

5. P.23 “as was in … condition” should be “as was the case in the … condition”

6. P.23 It should be “only when one generates … but not when one is”; on p.26 it should also be “one may .. with one’s worldview”; p.27 “If one experiences … one may want to … keep one’s assumptions”; p.28 “one can…one declares that one does not believe”

7. P.25 “not exclusively for the previous explanation” should be “not mutually exclusive with the previous explanation”

8. P.26: “a crucial role may play a number of factors” should be “a number of factors may play a crucial role”

9. P.27 “yet take advantage of misinformation” should be “yet rely [or: make use] of misinformation”; “failure to the retraction retrieval” should be “failure of retraction retrieval”

10. P.27 “by, for example, by”

11. P.27: Re “there must be some explanation for certain events", the authors may find this interesting: https://doi.org/10.1111/j.1475-682X.2009.00280.x

12. P.28 “exposed to some problem-solving process” should be “involved in some problem-solving process”

13. P.30 “under retractions’ highly credible source condition” – please rephrase (e.g., “under conditions where a retraction came from a highly credible source”)

14. P.30 either say “demonstrate consistent inoculation efficacy” or “demonstrate a consistent inoculation effect”

15. P.31 “exploring the inoculation” should be “exploring inoculation”

16. P.31 “impact of inoculation on misinformation” should be “impact of inoculation on misinformation reliance”

Reviewer #3: This manuscript presents one study testing the effects of the inoculation technique on the continued influence effect (CIE), while also addressing the role that source characteristics like credibility, trustworthiness, and expertise play on the effectiveness of inoculation (and of retractions of misinformation, in general). The topic of the manuscript is a timely and relevant one. When reading the manuscript, I liked that the authors addressed the issue of misinformation correction by bringing together solid theoretical frameworks – those involved in the CIE and the inoculation theory. I also liked the Discussion section; I specifically liked that the authors tried to integrate the different theories in light of their results and put forward an alternative conceptual proposal for the CIE.

In general, I feel positive towards the publication of this work. But I have some comments that I’d like the authors to address before the manuscript is accepted for publication. Most of these comments have to do with the source characteristics that were manipulated, and one final comment refers to integration of the inoculation technique with relevant research regarding the detection of fake news and misinformation. I list my comments below:

1) Given that the manipulation of source characteristics is central to the study, I was surprised that those characteristics were not even briefly discussed in the introduction section. As such, the reader is left wondering why different effects of trustworthiness and expertise should be expected, or why a credible source should be more efficient in what comes to reducing/eliminating the effects of misinformation. The characteristics of the source should also be discussed in terms of their relation to the mechanisms underlying the CIE, and the inoculation technique, in processual terms - how do they interfere with the processing of information and how can this be integrated with the mechanisms underlying the CIE and inoculation?

2) Related to point 1, the hypotheses are confusing, and some seem to mix simple main effects with qualification/interaction effects. For example, in H1, the authors say “Half of the participants had previously undergone the inoculation procedure. Given that retractions are more effective if their source is trusted but not necessarily expert (16,19,50), replication of existing reports can be expected (Hypothesis 1).” So, first it seems they will refer to the inoculation effect, which they don’t. Then, they talk about trustworthiness vs. expertise effects. And finally, they say “replication of existing reports can be expected (Hypothesis 1).” What does this mean? Which reports? Do the authors mean previous studies, such as Ecker and Antonio’s study? H2 is also confusing and unclear. The authors say “We also expected different vaccination effects depending on the reliability of the sources of the retractions. The inoculated individual could rely less on misinformation if the source of the correction was credible in both dimensions: trustworthiness and expertise (85), but not if it was solely expert or trustworthy (Hypothesis 2)”. Why is that so? Why should we expect these qualifications of the retraction effects (the authors never explain why they expect that corrections from sources will be accepted more if the source is both trustworthy and expert, rather than just having one characteristic? Why is that?), and why only for the inoculated participants? I can’t find the rational for the hypotheses in the information about source characteristics that is provided in the introduction (which is very little). The rational for H3 and H4 also needs to be better explained. I believe all the hypotheses would be clearer if the introduction gave the reader more information on source characteristics effects and their relation to correction of misinformation, the CIE, and inoculation.

3) I found the Methods section a little confusing, especially in what concerns the description of the materials and the procedure of the retraction scenarios. Regarding the operationalization of the different types of sources, it is difficult to understand exactly what it meant by expert, trustworthy, or credible sources. I know the materials are available in an appendix, but it would make the comprehension of the manipulations easier if one example could be given in the materials and procedure section.

4) Related to this, when examining the different scenarios, I felt that expert sources were somewhat different between the scenarios. As an example, in scenario 1, the expert source is “technicians employed at the municipal wastewater treatment plant”; in scenario 2, the expert source is “Evan's friend”. It seems to me that expert sources in scenario 1 align a lot more with the definition provided by the authors (“a source with access to information [not as a result of professionalism, but simply because of the possibility to access true information]”) than the expert source in scenario 2. Also, in scenario 5, why is a local newspaper trustworthy and not expert? Were the scenarios and the sources taken from previous research? To what extent can we be sure that sources were indeed interpreted as credible, expert, and trustworthy? And to what extent can we be sure that there were no differences in these interpretations between scenarios? Unless the manipulations were taken from previous research (it’s not clear, and if so, it should be clearly stated which scenario and which source operationalization came from which previous study) that had measures to make sure the sources are indeed interpreted as having the characteristics they’re intended to have, the possibility of differences between the scenarios should at least be discussed.

One note regarding the materials appendix, scenario 4 seem to be incomplete (the first message introducing the story is missing?)

5) Given the extension of the analyses and results presented, it would be easier to follow if the authors referred to the specific hypotheses stated in the introduction that each analysis and effects relate to.

6) Regarding the practical implications, I was not entirely convinced how one can implement the inoculation technique in day-to-day situations and in the general situations where misinformation is encountered. What comes to my mind is to have the kind of exercise the authors used to inoculate participants in media and information websites, as a way to make people aware of the prevalence of misinformation which in turn may make them more careful and detailed in their analysis and processing of the information they get. And this made me think of the work by Gordon Pennycook and his collaborators showing that propensity for analytical thinking is positively correlated with the ability to discern fake news from real news (e.g., Pennycook & Rand, 2019). I believe the manuscript would gain if the authors discussed the effects of inoculation also in light of this line of research trying to find ways to combat misinformation.

7) The full text should be proof read by a native English speaker, as there are some typos and some strange sentence formulations. This may be one of the reasons why some sections were confusing (e.g., the hypotheses, the methods).

7. PLOS authors have the option to publish the peer review history of their article (what does this mean?). If published, this will include your full peer review and any attached files.

Reviewer #1: No

Reviewer #3: No

---

## [Author Response · Author response to Decision Letter 1]

31 Mar 2022

Prof. Romuald Polczyk

Institute of Psychology, Jagiellonian University

6 Romana Ingardena Street, 30-060 Kraków, Poland

Tel. (+48) 12 663 24 36

Email: romuald.polczyk@uj.edu.pl

RE: Revised version of the article: “Vaccination against misinformation: The inoculation technique reduces the continued influence effect”

Dear Editors,

Please find enclosed the revised version of our article. Below we list all changes that have been made:

Major changes:

1. We have added a paragraph about the source characteristics and their relation to the CIE and inoculation mechanisms (p. 10/11).

2. We have rebuilt the hypothesis section, linking it to the newly added paragraph mentioned above (p. 11/12) and adding many more explanations.

3. We have added a new “Power analysis” section, in which we have reworked the fragment about power analysis (p. 13). We also have added a fragment about the limitations of achieved power in the discussion (p. 33).

4. We have expanded the fragment about the practical implications of our results in improving inoculation interventions (p. 32/33).

5. We have corrected some linguistic mistakes thorough the text.

6. Aside from the research mentioned by reviewers, we have added some new research references in the introduction and in the discussion sections:

a. https://doi.org/10.1111/j.1475-682X.2009.00280.x

b. https://doi.org/10.1080/10463280802643640

c. https://doi.org/10.1111/j.1559-1816.2010.00620.x

d. https://doi.org/10.1016/j.cognition.2018.06.011

e. https://doi.org/10.1038/s44159-021-00006-y

f. Szpitalak, M. (2015). W kierunku poprawy jakości zeznań świadków. Pozytywne i negatywne następstwa ostrzegania o dezinformacji. Wydawnictwo Uniwersytetu Jagiellońskiego.

We have also tried to address to all comments and recommendations made by the reviewers. Here is the list of the specific changes and corrections:

Reviewer 1:

1. We have added a new “Power analysis” section in which we have reworked the fragment about power analysis (p. 13). We hope that it is more adequate now. Also, we have added a fragment about the limitations of achieved power in the discussion (p. 33).

2. We have changed “behavioral memory statements” to “behavioral manifestations of memory” (p. 2, Abstract).

3. Unfortunately, we did not find any oddities in the format of “Guillory & Geraci”, therefore we did not make any corrections.

4. We have corrected “most likely be an effect” to “most likely an effect” (p. 24).

5. We have corrected “as was in … condition” to “as was the case in the … condition” (p. 25).

6. We have corrected:

a. “only when one generates … but not when they are” to “only when one generates … but not when one is” (p. 25);

b. “one may ... with their worldview” to “one may ... with one’s worldview” (p. 28);

c. “If one experiences … they may want to … keep their assumptions” to “If one experiences … one may want to … keep one’s assumptions” (p. 29);

d. “one can … they declare that they don’t believe” to “one can…one declares that one does not believe” (p. 30).

7. We have corrected “not exclusively for the previous explanation” to “not mutually exclusive with the previous explanation” (p. 27).

8. We corrected “a crucial role may play a number of factors” to “a number of factors may play a crucial role” (p. 28).

9. We have corrected:

a. “yet take advantage of misinformation” to “yet make use of misinformation” (p. 29);

b. “failure to the retraction retrieval” to “failure of retraction retrieval” (p. 29).

10. We have corrected “by, for example, by” to “for example, by”.

11. We have added https://doi.org/10.1111/j.1475-682X.2009.00280.x to the discussion, as suggested (p. 30).

12. We have corrected “exposed to some problem-solving process” to “involved in some problem-solving process” (p. 30).

13. We have rephrased “under retractions’ highly credible source condition” to “under conditions where a retraction came from a highly credible source” (p. 32).

14. We have changed “inoculation efficacy” to “inoculation effectiveness” (p. 33).

15. We have corrected “exploring the inoculation” to “exploring inoculation” (p. 34).

16. We have corrected “impact of inoculation on misinformation” to “impact of inoculation on misinformation reliance” (p. 34).

Reviewer 3

1. We have added a paragraph about the source characteristics and their relation to the CIE and inoculation mechanisms (p. 10/11).

2. We have rebuilt the hypothesis section, linking it to the newly added paragraph mentioned above (p. 11/12) and adding much more explanations. We hope that the hypotheses are now more understandable.

3. We have added a table with examples of certain sources from one of the scenarios (p. 15).

4. Unfortunately, it is true that we have no way of objectively assessing whether and to what extent may be the differences between the same source conditions in different scenarios. The best way to standardize these examples would be to perform a pilot study; however, for obvious reasons, we cannot do this. Thus, we made the decision to cover this issue on p. 33.

Also, Scenario 4 in the Appendix seems to be complete (the scenario was taken from Ecker & Antonio (2021), where it starts similarly).

5. We have added references to the specific hypotheses in the Results sections

a. H1 – p. 19

b. H2 – p. 19

c. H3 – p. 19

d. H4 – p. 20/21

e. H5 – p. 21/22

6. We have expanded the fragment about the practical implications of our results in improving inoculation interventions (p. 32/33). Although we could not add the proposed text, because our study did not include analytical thinking, we added it on p. 33.

7. Although we could not add the proposed Pennycook and Rand text in this excerpt, because our study did not include analytical thinking, we have added it to p. 32/33.

Yours sincerely

Romuald Polczyk

---

## [Editor Report · Decision Letter 2]

11 Apr 2022

Vaccination against misinformation: The inoculation technique reduces the continued influence effect

PONE-D-21-17666R2

Dear Dr. Polczyk

We’re pleased to inform you that your manuscript has been judged scientifically suitable for publication and will be formally accepted for publication once it meets all outstanding technical requirements.

Kind regards,

Margarida Vaz Garrido

Academic Editor

PLOS ONE

Additional Editor Comments (optional):

The authors did a great job in addressing most of the final comments. I think this paper is now greatly improved and ready for publication. I wish the authors much success in their future research endeavors.
---

## [Editor Report · Acceptance letter]

19 Apr 2022

PONE-D-21-17666R2 

Vaccination against misinformation: The inoculation technique reduces the continued influence effect 

Dear Dr. Polczyk:

I'm pleased to inform you that your manuscript has been deemed suitable for publication in PLOS ONE. Congratulations! Your manuscript is now with our production department. 

Kind regards, 

on behalf of

Dr. Margarida Vaz Garrido 

Academic Editor

PLOS ONE